# Genome-Wide Epistasis Study of Cerebrospinal Fluid Hyperphosphorylated Tau in ADNI Cohort

**DOI:** 10.3390/genes14071322

**Published:** 2023-06-23

**Authors:** Dandan Chen, Jin Li, Hongwei Liu, Xiaolong Liu, Chenghao Zhang, Haoran Luo, Yiming Wei, Yang Xi, Hong Liang, Qiushi Zhang

**Affiliations:** 1College of Intelligent Systems Science and Engineering, Harbin Engineering University, Harbin 150001, China; denise0620@163.com (D.C.); lijin@hrbeu.edu.cn (J.L.);; 2School of Automation Engineering, Northeast Electric Power University, Jilin 132012, China; 3School of Computer Science, Northeast Electric Power University, Jilin 132012, China

**Keywords:** Alzheimer’s disease, epistasis, hyperphosphorylated tau (P-tau), PPI, ADNI

## Abstract

Alzheimer’s disease (AD) is the main cause of dementia worldwide, and the genetic mechanism of which is not yet fully understood. Much evidence has accumulated over the past decade to suggest that after the first large-scale genome-wide association studies (GWAS) were conducted, the problem of “missing heritability” in AD is still a great challenge. Epistasis has been considered as one of the main causes of “missing heritability” in AD, which has been largely ignored in human genetics. The focus of current genome-wide epistasis studies is usually on single nucleotide polymorphisms (SNPs) that have significant individual effects, and the amount of heritability explained by which was very low. Moreover, AD is characterized by progressive cognitive decline and neuronal damage, and some studies have suggested that hyperphosphorylated tau (P-tau) mediates neuronal death by inducing necroptosis and inflammation in AD. Therefore, this study focused on identifying epistasis between two-marker interactions at marginal main effects across the whole genome using cerebrospinal fluid (CSF) P-tau as quantitative trait (QT). We sought to detect interactions between SNPs in a multi-GPU based linear regression method by using age, gender, and clinical diagnostic status (cds) as covariates. We then used the STRING online tool to perform the PPI network and identify two-marker epistasis at the level of gene–gene interaction. A total of 758 SNP pairs were found to be statistically significant. Particularly, between the marginal main effect SNP pairs, highly significant SNP–SNP interactions were identified, which explained a relatively high variance at the P-tau level. In addition, 331 AD-related genes were identified, 10 gene–gene interaction pairs were replicated in the PPI network. The identified gene-gene interactions and genes showed associations with AD in terms of neuroinflammation and neurodegeneration, neuronal cells activation and brain development, thereby leading to cognitive decline in AD, which is indirectly associated with the P-tau pathological feature of AD and in turn supports the results of this study. Thus, the results of our study might be beneficial for explaining part of the “missing heritability” of AD.

## 1. Introduction

Alzheimer’s disease (AD) is an insidious neurodegenerative disorder. The currently available therapies do not slow disease progression, provides short-term symptomatic relief only [1,2]. Genome-wide association studies (GWAS) and related techniques are gradually discovering variants of causal gene variants that contribute to complex human diseases. However, after years of GWAS efforts by countless researchers, these findings can explain only a small fraction of the heritability and the genetic factors of many human diseases and traits failed to be discovered, the so-called “missing heritability” [3,4]. Many research suggest that “missing heritability” in AD remains extensive with an estimated 25% of phenotypic variance unexplained by known variants, which may be explained by epistasis [5,6,7]. A single nucleotide polymorphism (SNP) is defined as single nucleotide alteration in a DNA sequence among individuals [8,9]. As a major drawback, in common GWAS, the focus is usually on SNPs that have significant individual effects [10,11]. Epistasis is the phenomenon about the interaction alleles of different loci when expressing a certain phenotype, and it cannot be attributed to the additive combination of effects corresponding to the individual loci. If the effect of one variant affecting a complex trait depends on the genotype of a second variant affecting that trait, epistasis will occurs [12]. To be specfic, epistasis leads to complex phenotypic effects, in which the effect of one locus is masked by the effects on another locus or the joint effects of two SNPs may be significant whereas they are ineffective separately [13,14]. Therefore, epistasis detection is expected to explain the “missing heritability” of many complex diseases such as AD, diabetes, and hypertension [15,16,17].

As the number of interactions grows exponentially with the number of variants, computational limitation is a bottleneck [18]. Most methods of epistasis detection choose to refrain from the brute force search in the SNP–SNP interaction space and try to reduce computational burden using dimensionality reduction screening and priori knowledge [19,20]. However, using more subjective priori knowledge or random factors for dimension reduction search will lead to signal loss because the risk of epistatic interaction is unknown [21]. Moreover, most methods of epistasis detection have been designed for case-control tasks over the past decade, with few on quantitative traits (QT) [19]. Compared to case–control status, QT has increased statistical significance and could better track AD progression [22]. Therefore, mining more potential loci by QT, which are implicated in AD without dropping signals across the whole genome, is urgently needed.

Emerging data have suggested that the prevailing amyloid cascade hypothesis is insufficient to explain many aspects of AD pathogenesis, neuroinflammation also plays an important role in the pathogenesis of AD [2,23]. AD is characterized by extracellular amyloid-β (Aβ) peptides in cortical Aβ plaques, intracellular phosphorylated tau protein as neurofibrillary tangles, and neuronal as well as axonal degeneration. These key hallmarks of AD can be measured in vivo with positron emission tomography (PET) imaging and biofluid markers including plasma and CSF assays [24]. CSF tau, P-tau, and A$\beta_{42}$ are established biomarkers for AD and have been widely used as QT for genetic analyse [25]. Furthermore, accumulated P-tau may be the primary contributor to neurodegeneration during AD, and neuroinflammation is a central mechanism involved in neurodegeneration as observed in AD and might play a critical role in inducing neurodegeneration [26,27,28]. Moreover, P-tau, one of the three candidate CSF QT, has been widely studied in the Alzheimer’s Disease Neuroimaging Initiative (ADNI) cohort. For example, heterogeneity in p-tau species carries predictive power in the identification of disease severity in incipient AD [29]. P-tau might start neurodegenerative processes and are necessary for cognitive decline [30]. Tau pathology is an initiating factor in sporadic AD [31]. Increase of p181tau levels might predict preclinical AD in cognitively normal elderly [32].

Therefore, CSF P-tau was used as a QT to advance statistical power and biological interpretation in this study. Then, we performed genome-wide epistasis detection in ADNI cohort based on a multi-GPU method, which has a better detection power outperform other competitive approaches.

## 2. Materials and Methods

### 2.1. Genotyping Data and Subjects Processing

Data used in this study were obtained from the ADNI database. The ADNI is a longitudinal multi-center study designed to be used for the early detection and tracking of AD, which was founded in 2004 under the leadership of Dr. Michael W. Weine and supported by the Foundation for the National Institutes of Health ($27 million) and the National Institute on Aging ($40 million). The primary goal of ADNI was developing biomarkers as outcome measures for clinical trials, examining biomarkers in earlier stages of the disease, and developing biomarkers as predictors of cognitive decline, etc. The SNP data were collected from the Illumina 2.5M array and the Illumina OmniQuad array including ADNI-1, ADNI-GO, and ADNI-2 cohorts, which can be downloaded from the LONI website (https://adni.loni.usc.edu, accessed on 19 June 2023). A total of 687,414 SNPs were involved in the study. To get pure SNP data, genetic analysis tool PLINK v1.90 was used to filter the SNPs according to the following quality control (QC) criteria: (1) SNPs on chromosome 1–22; (2) minimum call rate for SNPs and subjects ≥ 95%; (3) minimum allele frequencies (MAF) ≥ 5%; (4) Hardy-Weinberg equilibrium (HWE) test *p* ≥ 10${}^{-6}$. After the QC, a total of 563,980 SNPs participated in this study.

Subjects were checked by the following QC flow: (1) call rate per subjects ≥ 90%; (2) gender check; (3) identity check. Then, EIGENSTRAT was used to perform the population stratification analysis [33]. The population stratification analysis yielded 89 subjects who were non-Hispanic Caucasians. These 89 participants were excluded from the analysis. Finally, 1079 subjects passed the QC.

CSF P-tau phenotype was used as QT in this study. The QC criteria of phenotype was based on two principles: baseline consistency principle and normal distribution principle. Out of the 1079 subjects retained after the QC, 860 subjects had both genotype data and phenotype (CSF P-tau). These subjects (*N* = 860) including 201 cognitive normal cognition (CN), 84 significant memory impairment (SMC), 251 early mild cognitive impairment (EMCI), 209 late mild cognitive impairment (LMCI), and 115 AD subjects.

Overall, 860 valid P-tau of CSF subjects and 563,980 remained for the subsequent genome-wide SNP-SNP interaction analysis.

### 2.2. Genome-Wide SNP-SNP Interaction Analysis

In this study, including as covariates in the liner regression analysis such as age, gender and clinical diagnostic status (cds), we consider the linear regression model of additive main effect of two SNPs,
(1)$$L_{1,2}=\alpha_{0}+\alpha_{1}\times SNP_{1}+\alpha_{2}\times SNP_{2}+age+gender+cds+\varepsilon_{i}$$
where $\alpha_{0},\alpha_{1}$ and $\alpha_{2}$ are regression coefficients; $\varepsilon_{i}$ is a residual that follows a normal distribution with mean zero and variance $\sigma_{2}$. Therefore, the sum of both additive main effect and SNP${}_{1}$-SNP${}_{2}$ interaction is then given by
(2)$$L_{1,2}{}^{S}=\alpha_{0}+\alpha_{1}\times SNP_{1}+\alpha_{2}\times SNP_{2}+\alpha_{1,2}\times SNP_{1}\times SNP_{2}+age+gender+cds+\varepsilon_{i}$$
where $\alpha_{0}$, $\alpha_{1}$, $\alpha_{2}$ and $\alpha_{1,2}$ are regression coefficients. We set Y = Y${}_{1}$ Y${}_{2}$ … Y${}_{n}$, where n is the number of subjects in the sample, then the signal of SNPs is set in the form of S${}_{i}$ = S${}_{1j}$ S${}_{2j}$ … S${}_{nj}$, where j = 1 or 2; S${}_{ij}$ is the genotype of the allele on the SNP${}_{j}$ of the *i*th subject; S${}_{ij}$ = 0, 1 or 2. For each SNP-SNP interaction pair, the interaction effect was evaluated by two linear regression models according to the CSF P-tau quantitative trait. In practice, we test the significance of the interaction terms using an F test, then the *p*-value would be calculated.

### 2.3. Bioinformatics Analyses

To further explain the biological functions of SNP-SNP interaction pairs with significant interactions, all SNPs were mapped to the corresponding genes according to position based on the Homo sapiens genome assembly GRCh37 (hg19), and SNPs not located within the gene region were mapped to nearby genes by position offset of 100 kb. Moreover, the gene databases National Center for Biotechnology Information Phenotype-Genotype Integrator (NCBI PheGenI) was used to analyze the association of the candidate genes with phenotype trait, and Reactome 2022 was used to conduct pathway analysis for discover associated biological processes. For the gene-gene interaction pairs after mapping from the SNP level, functional enrichment processes were performed by PPI network enrichment analysis through the STRING database.

## 3. Results

### 3.1. SNP-SNP Interaction Results

In this study, a genome-wide SNP-SNP interaction detection using CSF P-tau as the intermediate quantitative phenotype was implemented. With the assistance of the GEEpiQt tool [34], we completely detected all SNP-SNP interaction pairs with significant interaction across the whole genome. According to the set *p*-value criteria, 758 SNP interaction pairs passed the significance requirement. After the GWAS analysis was conducted on the SNPs of 758 interaction pairs, all the results indicated that the statistical significance of the interaction effect was much higher than the main effect. The interaction effects and main effects of 758 significant SNP-SNP interaction pairs are shown in Figure 1.

To further confirm the association of SNP interactions with quantitative phenotypes, the explained variance of genetic epistasis of CSF P-tau was calculated by IBM SPSS 24.0. Two general linear models were used, with age, sex and disease diagnosis status added as covariates, and interaction terms were added to one model for calculating the main effect and interaction effect on phenotypic explanatory rate separately. The R square of the interaction terms and the additive terms are shown in Figure 2. The top 10 R-square of SNP-SNP interaction pairs and results of post hoc analysis on P-tau level are seen in Table 1. Age, gender, and cds accounted for 9.3% of variance on the P-tau level. Moreover, Table 1 gives the proportion of additional variance in P-tau level explained by the combined main effect and interaction effect of SNP${}_{1}$ and SNP${}_{2}$ after accounting for age, gender, cds, SNP${}_{1}$ and SNP${}_{2}$. The percentages of each interaction pair are as follows. For rs2291948 (*APOOP5*)—rs2619171, the interaction term accounted for 5.6% of variance, and the main effects accounted for 0.1% of variance (5.7% combined). For rs17069204 (*SEC63*)—rs4983187 (*LINC02588*), the interaction term accounted for 5.5% of variance, and the main effects accounted for 0.8% of variance (6.3% combined). For rs6882813—rs17416058, the interaction term accounted for 5.5% of variance, and the main effects accounted for 0.7% of variance (6.2% combined). For rs129600 (*PPARA*)—rs6602151 (*RSU1*), the interaction term accounted for 5.4% of variance, and the main effects accounted for 0.5% of variance (5.9% combined). For rs6796502 (*PRSS42P*)—rs6999890 (*SLC45A4*), the interaction term accounted for 5.3% of variance, and the main effects accounted for 0.4% of variance (5.7% combined). For rs1412839 (*PDPN*)—rs2397718, the interaction term accounted for 5.3% of variance, and the main effects accounted for 0.5% of variance (5.8% combined). For rs2219872 (*GRIP1*)—rs2647911 (*C12orf66*), the interaction term accounted for 5.2% of variance, and the main effects accounted for 1.3% of variance (6.5% combined). For rs9320250 (*OSTM1*)—rs4983187 (*LINC02588*), the interaction term accounted for 5.2% of variance, and the main effects accounted for 1.1% of variance (6.3% combined). For rs10802434 (*SCCPDH*)—rs12470444 (*NRP2*), the interaction term accounted for 5.2% of variance, and the main effects accounted for 0.6% of variance (5.8% combined). For rs2487643 (*PDPN*)—rs2397718, the interaction term accounted for 5.2% of variance, and the main effects accounted for 0.6% of variance (5.8% combined).

### 3.2. Functional Annotations for Significant Interaction Pairs

A total of 1161 SNPs were mapped onto 578 genes. Then, gene set enrichment analysis was performed based on HDSigDB Human 2021 in Enrichr. The results of enrichment analysis indicate that 331 genes have been shown associated with AD, and the number of unconfirmed AD-related genes is 247. In order to make more sensible explanations at the gene level in subsequent study, these gene pairs were categorized according to the relationship of genes with AD on both sides. The interactions were classified into three categories: both genes in each pair are AD-related, only one gene in each pair is AD-related, and none gene in each pair is associated with AD, as shown in Figure 3.

Moreover, the gene databases National Center for Biotechnology Information Phenotype-Genotype Integrator (NCBI PhenGenI) and Reactome 2022 were used to analyze the association of the detected genes with CSF P-tau quantitative traits. Pathway enrichment analysis of genes mapped by the identified SNPs based on their enrichment adjusted *p*-values is presented in Figure 4. The Reactome enrichment analysis showed that three of the top ten pathways were significant, Cell-Cell Communication, Cell Junction Organization and Neuronal System pathways. Among them, there is evidence that “Cell-Cell Communication” is associated with the abnormal accumulation of phosphorylated tau protein in Alzheimer’s disease. In the Neuronal System pathway, tau protein is normally involved in stabilizing microtubules in neurons, but in Alzheimer’s disease, it can become hyperphosphorylated and form aggregates called neurofibrillary tangles. These tangles can disrupt the transport of nutrients and other substances within neurons, which can further damage the pathways of the neuronal system. The results of the PhenGenI Association enrichment analysis were found significant in several diseases, such as Platelet Function Tests, Alzheimer Disease, and Stroke.

### 3.3. Potential Interactions via Protein-Protein Interaction Analysis

To investigate and validate the potential interactions additionally, we submitted 174 gene-gene interaction pairs to the STRING database for PPI enrichment analysis. All the 174 gene-gene interaction pairs were selected from the 1st relationship shows in Figure 3a and the 2nd relationship as shows in Figure 3b. Notably, 10 of the gene interaction pairs overlapped with PPI networks in the database, as shown in Figure 5, including: *SPSB1-EPHB1*, *HNRNPU-NEDD4L*, *MYT1L-NYAP2*, *GGCX-F13A1*, *LRP1B-PDE4D*, *RARB-NR3C1*, *CCL2-SEMA6D*, *ROBO1-TLE1*, *CSNK1A1-PTK7*, *MYO5B-PCDH15*.

## 4. Discussion

In this study, detection of genome-wide SNP-SNP interaction based on multi-GPU were performed. To our knowledge, this study is a highly comprehensive epistatic study of QT at the P-tau level. A total of 758 SNP-SNP pairs were found to be statistically significant, and highly significant SNP–SNP interactions were detected between the marginal main effect SNPs. In particular, the interaction effects were much higher than the main effects (Figure 1). As we expected, all identified interaction pairs explained a relatively high-level variance at the P-tau level (Figure 2 and Table 1), which could be helpful for explaining some part of the “missing heritability” of AD.

To identify potential genetic epistasis implicated in AD and obtain biologically meaningful explanations at gene-gene interaction level, 174 gene-gene interaction pairs, which from 1st relationship (Figure 3a) and 2nd relationship (Figure 3b) were submitted to the STRING database to perform the PPI enrichment analysis. As shown in Figure 5, the PPI sub-network containing 20 genes and 33 gene-gene interactions were identified. As a result, ten gene-gene interaction pairs overlapped with the PPI network need further discussion.

*CSNK1A1* is a casein kinase which is involved in the phosphorylation state of tau [35]. Protein tyrosine kinase 7(*PTK7*) is a regulator of Wnt signaling pathways [36]. Wnt signaling is deregulated in AD, which could contribute to synapse degeneration and cognitive decline. This deficiency in Wnt signaling may further exacerbate tau hyperphosphorylation [37]. Therefore, the *CSNK1A1* and *PTK7* interaction shows strong associations with tau phosphorylation.

The *SPRY* domain-containing *SOCS* box protein 1 (*SPSB1*) is involved in the development of AD through nitric oxide (NO) pathways. To be specific, NO pathways contribute to pathogenesis of neurodegeneration in AD and other neurodegenerative dementias by involving in neuroinflammation, while SPSB1 negatively control NO production and limit cellular toxicity [38]. Ephrin type-B receptor 1 (*EphB1*) is upregulated in injured motor neurons, and then activated astrocytes [39]. Furthermore, activated astrocytes mediate neuroinflammation and neurodegeneration. Neuroinflammation induces neurodegeneration and the processes involved in neurodegeneration augments neuroinflammation [26].

Neurodegeneration is mediated by inflammatory and neurotoxic mediators such as chemokine (C-C motif) ligand 2 (*CCL2*), *CCL5*, tumor necrosis factor-alpha (TNF-α) and interleukin-6 (*IL-6*) etc. The increased level of these mediators including *CCL2* lead to neurodegeneration and neuronal death in neurodegenerative diseases [26,40]. These inflammatory and neurotoxic mediators directly or indirectly through glial cells and inflammatory cells affect neuronal survival and induce neurodegeneration [26]. Moreover, *CCL2* is implicated in the pathways recruiting microglia and the development of P-tau pathology, and might be related to reducing neuroinflammation [41]. *SEMA6D* is a regulator of microglial phagocytosis and inflammatory cytokine (TNFα, IL-6 etc.) release in a TREM2-dependent manner [42,43]. Moreover, microglial phagocytosis is a disease-associated process emerging from AD genetics [44]. Excessive microglial phagocytosis of synapses can be observed in AD, leading to significant synapse loss and memory impairment [45]. And lack of microglial phagocytosis can exacerbate pathology of AD and induce memory impairment [46]. Microglia are also major players in neuroinflammation [23].

*RP1B* belonged to the low-density lipoprotein (LDL) receptor family, and several members of the LDL family have been implicated in cellular processes relevant to neurodegeneration, including tau uptake et al. Enhanced *LRP1B* activity can protect against the pathogenesis of AD and cognitive decline in old age. Inhibition of phosphodiesterase 4D (*PDE4D*) activity can enhance phosphorylation of tau.

*MYT1L* is a critical mediator of directly converting human brain vascular pericytes (HBVPs) into cholinergic neuronal cells, and the cholinergic deficit is thought to underlie progressed cognitive decline in AD. Neuronal tyrosine-phosphorylated phosphoinositide-3-kinase adapter 2 (*NYAP2*) is involved in remodeling of actin cytoskeleton [47]. Actin cytoskeleton has been described as an underlying factor of synaptic failure in AD, which could contribute to AD pathology [48].

Regulation of *HNRNPU* expression ameliorates impairments of learning and memory abilities in an AD rat model. *NEDD4L* is identified as potential nuclear enriched abundant transcript 1 (*NEAT1*) interaction proteins. upregulated NEAT1 can give rise to the amyloid accumulation and cognitive decline in AD.

In summary, *CSNK1A1-PTK7* interaction and *PDE4D* gene shows strong associations with P-tau, which is directly associated with pathogenesis of AD. Two pairs of gene-gene interactions show strong associations with AD in terms of neuroinflammation and neurodegeneration: *SPSB1-EPHB1*, *CCL2-SEMA6D*. *MYT1L* gene, *LRP1B* gene and *NEDD4L* gene show strong associations with AD, and leading to cognitive decline in AD. *LRP1B* gene is also associated with pathogenesis of AD. *HNRNPU* gene exerts its effects on learning and memory abilities in AD. *NYAP2* gene is involved in remodeling of actin cytoskeleton, which is indirectly associated with AD pathology. Two pairs of gene-gene interactions can affect the activation of neuronal cells and contribute to brain development: *RARB-NR3C1*, *ROBO1-TLE1* [49,50]. *MYO5B* gene and *GGCX* gene show strong associations with schizophrenia [51]. Neuroinflammation is well established in a subset of schizophrenia patients [52]. In addition, two genes have not yet been associated with AD pathology: *F13A1*, *PCDH15*, which warrant further investigation.

To our knowledge, neurodegeneration appears to be the biological mechanism most proximate to cognitive decline in AD [53]. Aβ and tau pathologies interact synergistically in the preclinical stages of AD, which contributing to faster neurodegeneration and cognitive decline [54,55]. Therefore, the identified gene-gene interactions and genes in the PPI network might be related to neuroinflammation and neurodegeneration, thereby leading to cognitive decline in AD, which is indirectly proves that accumulated P-tau may be the primary contributor to neurodegeneration during AD [27], and in turn supports the results of this study.

## 5. Conclusions

Aimed at performing genome-wide epistasis detection in the ADNI cohort, we used CSF P-tau as a QT. 331 AD-related gene were replicated, which are previously confirmed AD risk genes. We also replicated 10 findings in the PPI network, which are *SPSB1-EPHB1*, *HNRNPU-NEDD4L*, *MYT1L-NYAP2*, *GGCX-F13A1*, *LRP1B-PDE4D*, *RARB-NR3C1*, *CCL2-SEMA6D*, *ROBO1-TLE1*, *CSNK1A1- PTK7*, *MYO5B-PCDH15*. Moreover, 3 gene-gene pairs of interaction showed strong association with AD: *CSNK1A1-PTK7*, *SPSB1-EPHB1*, *CCL2-SEMA6D*. Interactions between *RARB* and *NR3C1*, between *ROBO1* and *TLE1* can affect the activation of neuronal cells and contribute to brain development. Our study also revealed two genes have not yet been associated with AD pathology: *F13A1*, *PCDH15*, which warrant further investigation. In summary, our results can provide useful clues to the aspect of inducing neuroinflammation and neurodegeneration, and show strong association with AD in terms of cognitive decline. Therefore, this study might open new avenues to complement common GWAS. Furthermore, our results may be replicated by considering other quantitative traits using different databases and methods to complement the PPI network. Biological interpretation is also a direction of future research.

## Figures and Tables

**Figure 1 genes-14-01322-f001:**
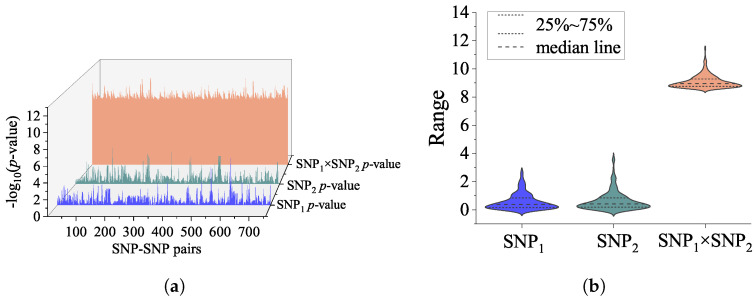
(**a**) The 3D waterfall plot reveals interaction effects and main effects of each SNP with −log${}_{10}$ (*p*-values). The number of significant SNP-SNP pairs was 758. For each interaction pair, the blue waterfall represents main effects of SNP${}_{1}$; the green waterfall represents main effects of SNP${}_{2}$; and the orange waterfall represents the interaction effects of SNP${}_{1}$–SNP${}_{2}$ pairs. (**b**) The violin plot shows the distribution state and probability density of SNP-SNP interaction effect, main effect of SNP${}_{1}$ and SNP${}_{2}$.

**Figure 2 genes-14-01322-f002:**
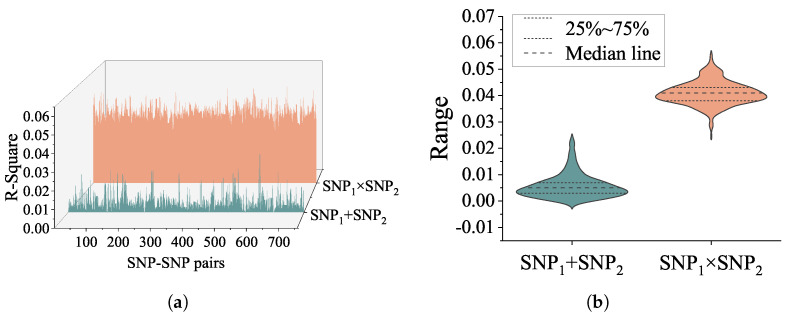
(**a**) The 3D waterfall plot reveals interaction and additive terms with R square in the linear regression model. The orange area represents the variance explained by interaction term on P-tau. The green area represents the variance explained by additive term on P-tau. (**b**) The violin plot shows the distribution state and probability density of interaction R square, additive terms of R square the two SNPs.

**Figure 3 genes-14-01322-f003:**
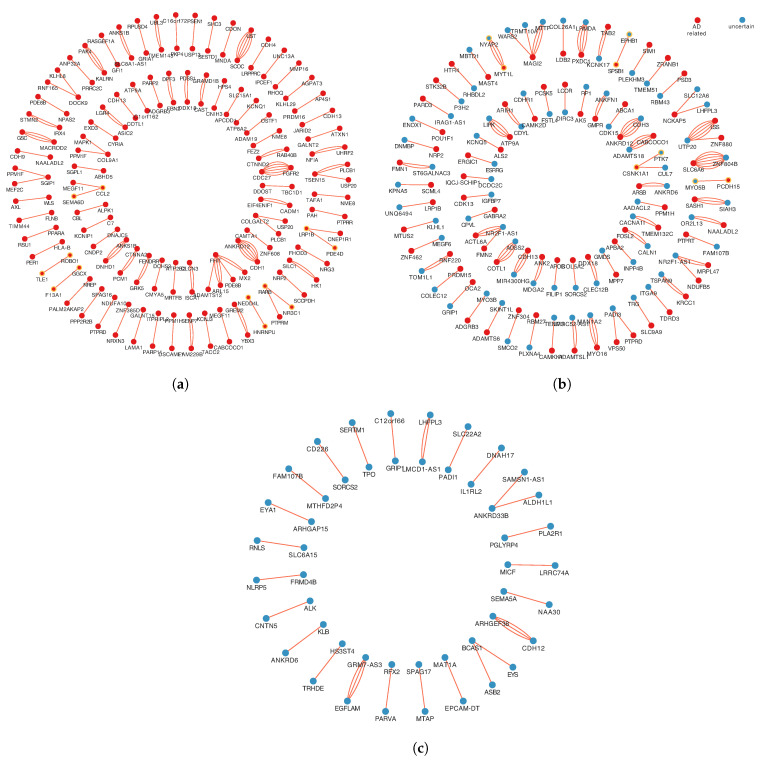
Three categories of different relationships with AD correlation in a gene-gene interaction pair. (**a**) The 1st relationship, include 95 gene pairs, both genes in each pair are AD-related. (**b**) The 2nd relationship, includes 79 gene pairs, only one gene in each pair is AD-related. (**c**) The 3rd relationship, includes 25 gene pairs, none gene in each pair is associated with AD. The red spots represent AD-related genes. The blue spots represent AD-unrelated genes. Multiple lines between two points represent multiple pairs of SNP mapped ont to the same pair of genes.

**Figure 4 genes-14-01322-f004:**
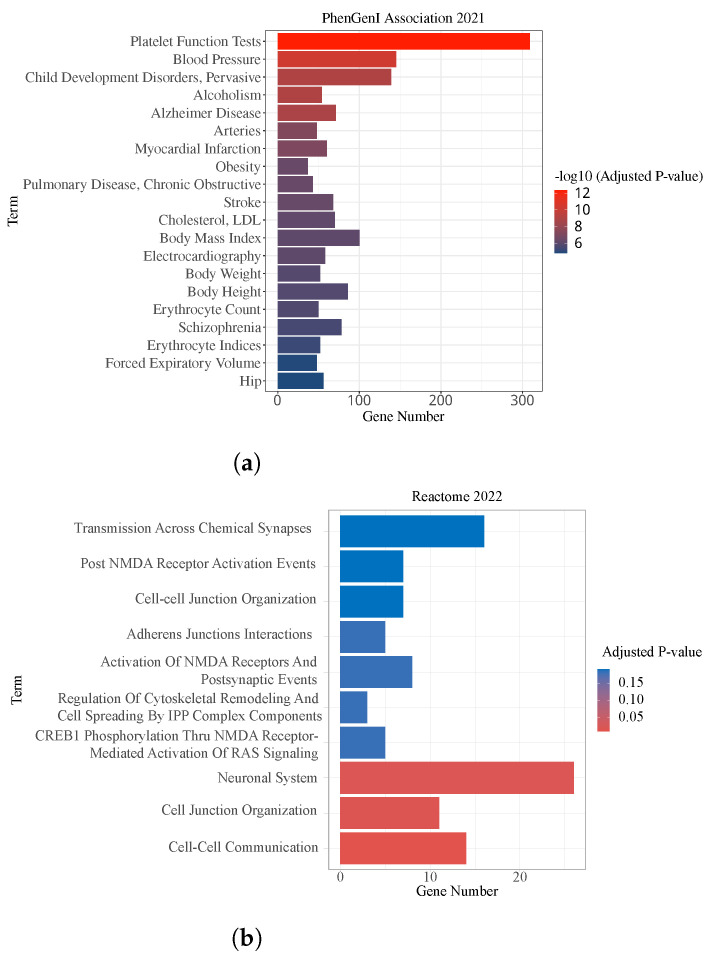
Significantly enriched pathways of studied genes. *x*-axis indicates the number of overlapped genes of related pathway, *y*-axis indicates significant pathways. The gradient of the color represents the level of significance. The red bars represent high significance. (**a**) Top 20 pathways identified by PhenGenI Association 2021 enrichment, −log10 adjusted *p*-value. (**b**) Top 10 pathways identified by Reactome 2022, adjusted *p*-value.

**Figure 5 genes-14-01322-f005:**
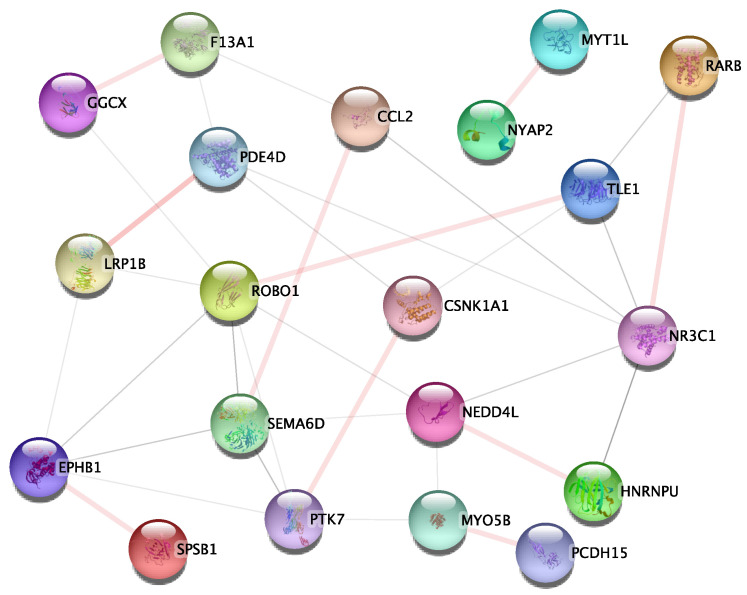
The PPI subnetwork of studied genes. The nodes and edges represent the proteins (genes) and their interactions, respectively. The PPI subnetwork contained 20 nodes and 33 edges. The light pink connections represent ten overlapped gene-gene interaction pairs with the PPI network.

**Table 1 genes-14-01322-t001:** Top10 R Square Of Snp-Snp Interaction Pairs.

NO	SNP ${}_{1}\times$ SNP ${}_{2}$	GENE	CHR	*p*-Value		Explained Variance (R Square)		
				GWAS	Interaction	Age + Gender + cdsr ${}^{1}$	SNP ${}_{1}$ + SNP ${}_{2}^{2}$	SNP ${}_{1}$ × SNP ${}_{2}^{3}$
1	rs2291948	*APOOP5*	16	0.963536	1.70 × 10${}^{-9}$	0.093	0.001	0.056
	rs2619171	*-*	15	0.911948				
2	rs17069204	* **SEC63** *	6	0.0222884	6.73 × 10${}^{-11}$	0.093	0.008	0.055
	rs4983187	*LINC02588*	14	0.592583				
3	rs6882813	*-*	5	0.420613	2.86 × 10${}^{-10}$	0.093	0.007	0.055
	rs17416058	*-*	11	0.212032				
4	rs129600	* **PPARA** *	22	0.635138	4.92 × 10${}^{-10}$	0.093	0.005	0.054
	rs6602151	* **RSU1** *	10	0.870792				
5	rs6796502	*PRSS42P*	3	0.676528	4.80 × 10${}^{-11}$	0.093	0.004	0.053
	rs6999890	*SLC45A4*	8	0.454329				
6	rs1412839	* **PDPN** *	1	0.251693	6.00 × 10${}^{-10}$	0.093	0.005	0.053
	rs2397718	*-*	5	0.785923				
7	rs2219872	*GRIP1*	12	0.016483	3.52 × 10${}^{-12}$	0.093	0.013	0.052
	rs2647911	*C12orf66*	12	0.292968				
8	rs9320250	*OSTM1*	6	0.0269661	4.37 × 10${}^{-10}$	0.093	0.011	0.052
	rs4983187	*LINC02588*	14	0.592583				
9	rs10802434	* **SCCPDH** *	1	0.873933	8.49 × 10${}^{-10}$	0.093	0.006	0.052
	rs12470444	* **NRP2** *	2	0.291239				
10	rs2487643	* **PDPN** *	1	0.231468	1.31 × 10${}^{-9}$	0.093	0.006	0.052
	rs2397718	*-*	5	0.785923				

${}^{1}$ Age + gender + cds: percent of variance in P-tau level explained by age, gender and cds. ${}^{2}$ SNP${}_{1}$ + SNP${}_{2}$: percent of additional variance in P-tau level explained by the combined main effect of SNP${}_{1}$ and SNP${}_{2}$ after accounting for age, gender and diagnosis. ${}^{3}$ SNP${}_{1}$ × SNP${}_{2}$: percent of additional variance in P-tau level explained by the interaction effect of SNP${}_{1}$ and SNP${}_{2}$ after accounting for age, gender, diagnosis, SNP${}_{1}$ and SNP${}_{2}$. The genes with bold italics in the table are AD-related genes.

## Data Availability

The data used in this project were funded by the Alzheimer’s Disease Neuroimaging Initiative (ADNI). The database is the (http://adni.loni.usc.edu/, accessed on 19 June 2023) of the Alzheimer’s disease neuroimaging database.

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
