# Peer review of "Genome-Wide Epistasis Study of Cerebrospinal Fluid Hyperphosphorylated Tau in ADNI Cohort"

_genes, 2023, doi:10.3390/genes14071322_

Round 1

Reviewer 1 Report

The manuscript presents only in silico work. Some results, or at least the main conclusions, should be verified by wet methods.

Minor issues

-Be careful with spaces, which are often missing between words or before parentheses.

-Species names should always be in italics.

-It is not p-Tau that initiates neurodegeneration, nor is it even clear that it is the tangles formed by this protein.

-Often the text of the figures is too small.

-Categories in Fig. 3 should be better explained.

-Table 1 is not explained in results section. Neither fig. 5

-Is not clear what lines indicate interaction pairs in fig.5

-Myt1l should be in capital letters.

-Last part of the discussion should be more related with AD.

-Conclusions are actually an abstract of the results

English should be checked by a native expert.

Author Response

Thank you for your comments concerning our manuscript entitled “Genome-wide epistasis studies of cerebrospinal fluid hyperphosphorylated tau in ADNI cohort” [Manuscript ID:genes-2456763]. These comments are of great help to improve the paper, which greatly enhances the readability of the paper. We have studied the comments carefully and have made corrections and changes accordingly. We hope this revised version is satisfactory (revised parts are marked in red).  Please see the attachment.

Reviewer 2 Report

The paper concerns Alzheimer's disease (genome wide methylation analysis of CSF in ADNI cohort). Included are 5 figures, 1 table SNPs, genes were mapped to the genome, and gene-gene interactions measured. The figures are of high resolution. PPI network generated as well. The authors could add citations to one of John Hardy AD gene papers. There are no major spelling/grammar errors. No further recommendations, only follow this minor correction.

Author Response

(The authors gave the same response as above.)

Reviewer 3 Report

This manuscript has several shortcomings, including:

1.     The title is difficult to understand.

a.     It is difficult to understand. How can genome-wide epistasis studies be conducted on cerebrospinal fluid hyperphosphorylated tau?

b.     ‘Epistasis’ and ‘ADNI’ need more explanation.

2.     The concept of “missing heritability” needs to be explained more fully.

3.     The term “the ultra-complex genetic drivers” of AD is both confusing and misleading. Autosomal dominant AD cannot be described as ‘ultra-complex’. Similarly, it is possible that some forms of AD have very little or no genetic component. Therefore, the text needs to be adjusted.

4.     The manuscript repeatedly uses the words ‘might’ and ‘may’, which suggests that the opposite (‘might not’ and ‘may not’) are equally likely. If, as I suspect,  this is not the case,  then the language used needs to be corrected.

5.     Line 50 – Shouldn’t it be emerging data ?

6.     Line 61 – how do SNPa ‘participate’ in a study?

7.     It is not completely clear how CSF-tau data was used. This needs to be corrected.

8.     Line 201 – I stand the use of the word ‘replicated’. This needs to be clarified.

Some phrases seem odd or out of place

Author Response

(The authors gave the same response as above.)

Round 2

Reviewer 1 Report

This version of the manuscript has been improved over the previous one. However, the major issues remain the same, since only the minor ones have been fixed.

English grammar and style should be checked by a native expert.